# Electrode Clustering and Bandpass Analysis of EEG Data for Gaze Estimation

**Ard Kastrati**[1]                                      akastrati@ethz.ch
**Martyna Beata Płomecka**[2]                            martyna.plomecka@uzh.ch
**Joël Küchler**[1]                                      kjoel@ethz.ch
**Nicolas Langer**[2]                                    n.langer@psychologie.uzh.ch
**Roger Wattenhofer**[1]                                 wattenhofer@ethz.ch
[1] *ETH Zurich, Switzerland*
[2] *University of Zurich, Switzerland*

## Abstract

In this study, we validate the findings of previously published papers, showing the feasibility of an Electroencephalography (EEG) based gaze estimation. Moreover, we extend previous research by demonstrating that with only a slight drop in model performance, we can significantly reduce the number of electrodes, indicating that a high-density, expensive EEG cap is not necessary for the purposes of EEG-based eye tracking. Using data-driven approaches, we establish which electrode clusters impact gaze estimation and how the different types of EEG data preprocessing affect the models' performance. Finally, we also inspect which recorded frequencies are most important for the defined tasks.

**Keywords:** EEG, Clustering, Deep Learning, Gaze Estimation, Bandpassing

## 1. Introduction

The ability to track eye movement patterns offers insights into the cognitive processes underlying a wide variety of human behaviour. Eye tracking allows researchers to recognize and quantify visual attention, fatigue and performance in various scientific studies (Eriksson and Papanikotopoulos (1997); Holmqvist et al. (2011); Liu and Heynderickx (2011)). Nowadays, infrared video-based eye trackers are the most common approach in research labs (Cornelissen et al. (2002)). This eye tracking technique uses infrared light to create a dark pupil and a corneal reflection to provide contrast in locating the center of the pupil (Holmqvist et al. (2011)). Although accurate, there are various limitations (Holmqvist et al. (2012)). Examples include individual differences in the contrast of the pupil and iris, time-consuming setup and calibration for each scanning session (Carter and Luke (2020)). Moreover, installing such a system involves setting up an optical path to align infrared light to the cornea without interference from the visual paradigm display.

Another line of research demonstrated the feasibility of Electroencephalography (EEG) and Electrooculography (EOG) signal decoding for gaze estimation purposes. EOG is often concurrently measured with EEG by using electrode pairs placed horizontally and/or vertically around the eye to record changes in electric potentials that originate from movements of the eye muscles (Martínez-Cerveró et al. (2020)). However, gaze estimation based on Electroencephalography (EEG) data was mainly overlooked in practice (Manabe et al.

(2015); Hládek et al. (2018)), even though the solution has been known for years. One potential reason is the necessity to obtain a rich enough set of data with concurrently recorded EEG data and infrared video-based eye tracking data serving as ground truth. In addition, this solution requires equipment and expertise for both EEG acquisition and eye tracking (Dimigen et al. (2011)). Moreover, the noisiness of EEG data poses an additional challenge leaving the research and development in the EEG-based gaze estimation area behind (Grech et al. (2008)). Nonetheless, with the rapid progress in Machine Learning and the increase in the collection of datasets, the EEG modality became more approachable and easier to process (Kastrati et al. (2021)).

The primary goal of this study is to investigate which spatio-spectral brain signal components are most relevant for decoding eye movement and understanding what is the minimal and best placement of the electrodes. More importantly, we show that good-performing models with high accuracy can be achieved even when the number of electrodes is significantly reduced compared to a high-density, 128-electrodes EEG cap. Additionally, we demonstrate that when using a standard pipeline for EEG preprocessing that includes independent component analysis (ICA) (Pion-Tonachini et al. (2019)), one can infer gaze direction also from electrodes placed in the occipital part of the head, albeit with much lower accuracy. Finally, we include an experimental analysis of the importance of different frequency intervals of the EEG signals for gaze direction.

## 2. Related Work

To date, only a few EEG studies attempted to estimate the actual eye gaze position on a computer screen, and those resulted in high inaccuracies (estimation error 15°) (Borji and Itti (2012)), and complicated analyses (Manabe et al. (2015)). Moreover, there are a number of limitations associated with EEG and specifically EOG electrodes, including a variety of metabolic activities over time and potential drifts (Martinsen and Grimnes (2011)). There has been some investigation into the traditional methods of supervised machine learning; for instance, (Bulling et al. (2010)) categorized the various directions and durations of saccades with a mean accuracy of 76.1%. Another study (Vidal et al. (2011)) developed an EOG feature-based approach that discriminated between saccades, smooth pursuits, and vestibulo-ocular reflex movement achieving quite good results, all of them being above 80%. Latest approaches in machine learning (ML) have shown promise for the development of more precise EEG/EOG-based eye tracking systems. For example, the EEGEyeNet (Kastrati et al. (2021)) benchmark, along with the rich dataset of simultaneous Electroencephalography and eye tracking recordings, has demonstrated promising results for further development of EEG-EOG-based gaze estimation using deep learning frameworks. Recently, (Wolf et al. (2022)) proposed a novel framework for time-series segmentation, creating ocular event detectors that rely solely on EEG data. This solution achieved state-of-the-art performance in ocular event detection across diverse eye tracking experiment paradigms, showing the high potential of EEG-based eye tracking solution, used not only as a complimentary modality but in some instances, it can also be beneficial when classic eye tracker hardware is unavailable.

## 3. Methods

### 3.1. Dataset, Benchmarking Tasks and Models

We use EEGEyeNet dataset and benchmark (Kastrati et al. (2021)) to run our studies. EEGEyeNet consists of synchronized EEG and eye tracking data and consists of collected from three different experimental paradigms. It also establishes a benchmark consisting of three tasks with an increasing level of difficulty:

- Left-Right task (LR): This is a binary classification task, where the goal is determining the direction of the subject's gaze along the horizontal axis.

- Direction task: The task is to regress the two target values, i.e., angle and amplitude of the relative change of the gaze position during the saccade.

- Absolute position task: The goal of the task is determining the absolute position of the subject's gaze in the screen, described in terms of XY-coordinates.

In each task, a window of 1 second (with sampling rate of 500Hz) of EEG data is given as an input sample, and the goal is to classify or regress (depending on the task) the target value. The data acquisition, models used, preprocessing methods, and the whole pipeline is explained in much more detail in the original work (Kastrati et al. (2021)).

### 3.2. EEG data preprocessing

In this manuscript, we use two types of EEG -data preprocessing. The first one, from now on referred to as "Minimal Preprocessing", includes algorithms implemented in the EEGlab plugin: to identify bad channels (`clean_rawdata`[1]), to reduce the noise (Zapline) and bandpass filtering (0.5-40Hz). Detected bad channels were automatically removed and later interpolated using a spherical spline interpolation. The details of the preprocessing pipeline can be found in the EEGEyeNet paper (Kastrati et al. (2021)). The second preprocessing type, i.e. "Maximal preprocessing", is the state-of-the-art preprocessing used for neuroscientific applications. In addition to the "Minimal Preprocessing" pipeline, it includes a Non-Brain artifactual source components removal based on the automatic classification result as provided by Independent Component Label (ICLabel) (Pion-Tonachini et al. (2019)) algorithm.

### 3.3. Gradient-Based Feature Importance

In order to indicate whether a feature (in our case an electrode) contributes significantly to the prediction, the gradient concerning the input can be computed (Simonyan et al. (2013)). This shows how much the model relies on the feature. In our case, we are interested in aggregating this score over an electrode to get its importance. First, the resulting gradients are normalized for each input separately. Next, all absolute gradient values of an electrode are summed up. If an electrode is left with a higher score comparatively, it indicates that it has a more significant contribution to the input.

---

1. http://sccn.ucsd.edu/wiki/Plugin_list_process

## 4. Electrode Clustering on Minimally Preprocessed Data

A dense, 128-electrode EEG cap is often infeasible to be used in practice. In this section, we show that most of the important information for eye movement is highly concentrated in the frontal electrodes.

### 4.1. Finding Important Electrodes

We use gradient-based methods to rank the most important electrodes. From the spatial distribution (topoplots) in Figure 1 we can see a clear symmetrical structure and that the most information used for decoding eye movement is in the frontal electrode. Interestingly, the topoplots show that the central electrode (Cz), which was the recording reference electrode also carries some relevant information. This can be explained, by the fact that the data were offline re-referenced to average reference and thus Cz was interpolated by all other electrodes. To produce the topoplots we used the average performance across all models and for each task presented in the benchmark (Kastrati et al. (2021)).

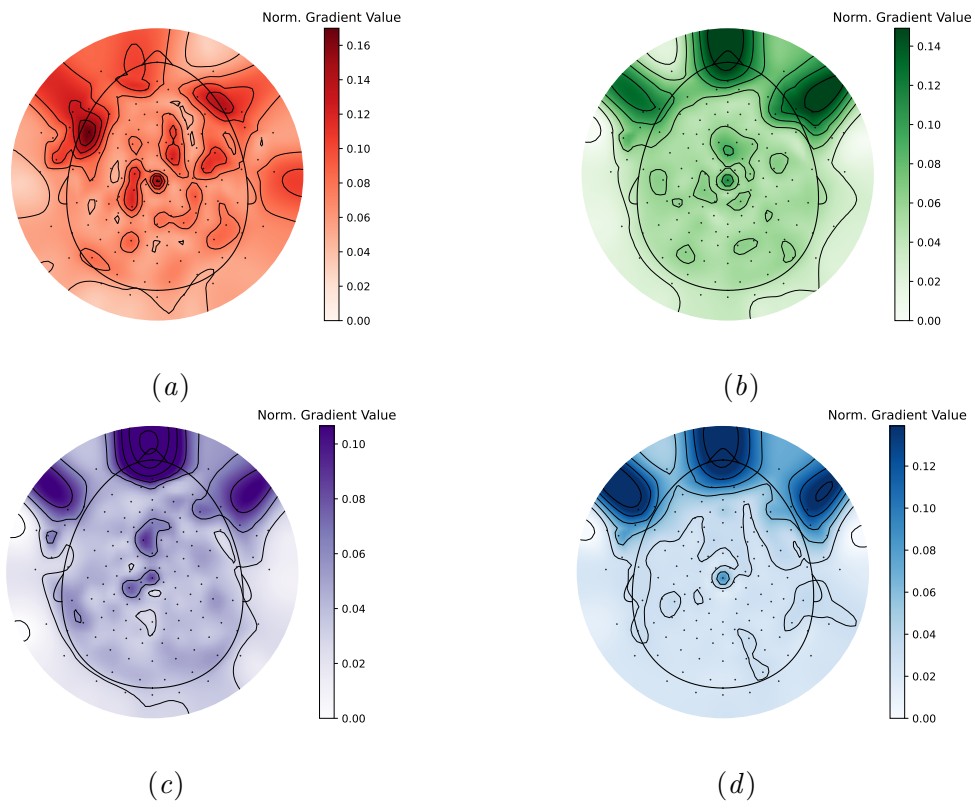

Figure 1: Topoplots for (a) LR Task, (b) Position Task, (c) Amplitude Task and (d) Angular Task. Colors of higher intensity represent a higher importance. The colors are shown on a logarithmic scale. The score is averaged over all five deep learning architectures presented in (Kastrati et al. (2021)).

## 4.2. Choice of Electrodes

Due to the symmetrical results on the topoplots, the most important electrodes are chosen as follows: first, all electrodes are ranked according to gradient-based analysis (see Figure 2 for the importance of each electrode). The best electrode, which is not yet in the cluster, is added together with its symmetrical counterpart. This method responds to the brain's natural symmetry and focuses on localized head areas that we expect to be more resilient and insightful than single isolated electrodes. This procedure is repeated until the loss stops increasing. With this method, we converged to 23 electrodes which are mainly on the frontal part (as can be observed in the topoplots and Figure 3). In order to compare models with each other and for convenience, it is ensured that the same configurations are used across all tasks and models.

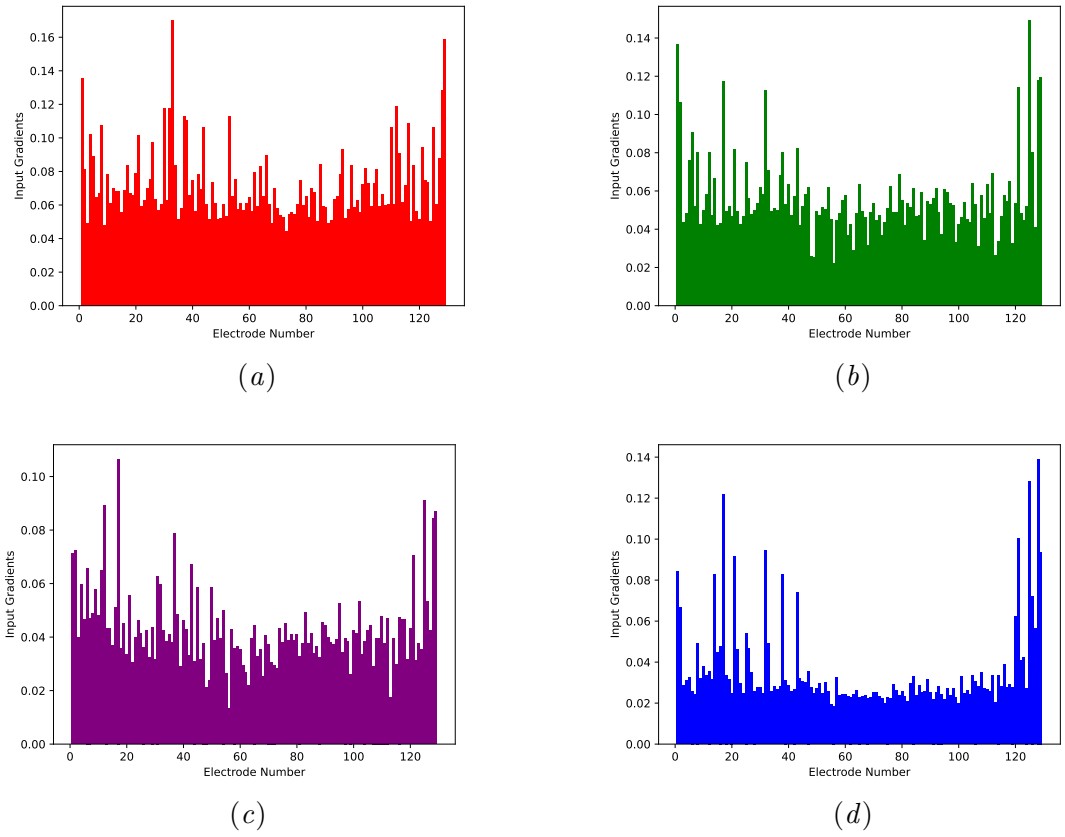

Figure 2: Gradient Based Feature Importance for (a) LR Task, (b) Position Task, (c) Amplitude Task and (d) Angular Task. The normalized gradient results for each electrode and each task from the experiments on the minimally preprocessed data.

### 4.3. Choice of Clusters

We observed that the best electrodes are of similar significance in all architectures for all the tasks and the accuracy. In addition, the choice of 23 electrodes achieves the same accuracy that one achieves with all 128 electrodes. This high accuracy encouraged us to train models on an even more reduced number of electrodes and this way we created several different and smaller clusters. We choose clusters of sizes 2, 3, 8, and 23. For all tasks and the models in EEGEyeNet benchmark, we converged to the same clusters shown below.

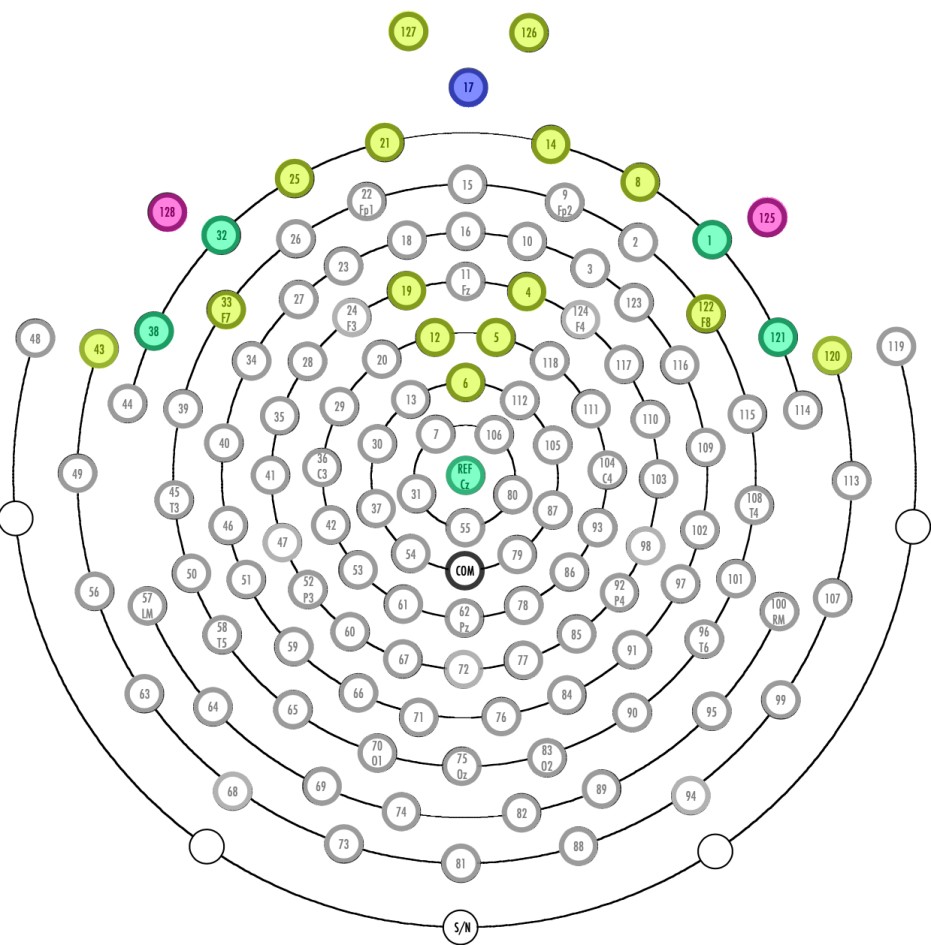

Figure 3: **Electrode Clustering Visualisation.** This figure shows the electrode placement. Colour-coded electrodes belong to a configuration. Pink nodes form the Top2 configuration. For Top3, the blue electrode is added. By combining Top3 with the teal electrodes, we get the Top8 configuration. The final 23 electrode composition consists of all coloured nodes.

| Name | Electrodes | Quantity | Colour Combination |
|---|---|---|---|
| Top2 | {125,128} | 2 | Pink |
| Top3 | Top2 ∪ {17} | 3 | Pink + Blue |
| Top8 | Top3 ∪ {1,32,38,121,129} | 8 | Pink + Blue + Teal |
| SideFronts | Top8 ∪ {4,5,6,8,12,14,19,21,25} ∪ {33,43,120,122,126,127} | 23 | Pink + Blue + Teal + Yellow |

Table 1: **Electrode Clustering.** Each electrode is given a colour as shown in Figure 3. The colours refer to the hues of the used electrodes.

## 4.4. Evaluation

| Cluster | Model | LR. (%) ↑ | Amp. (mm) ↓ | Ang. (rad) ↓ | Pos. (mm) ↓ |
|---|---|---|---|---|---|
| All (128) | InceptionTime | $97.07 \pm 0.2$ | $27.25 \pm 0.94$ | $0.36 \pm 0.06$ | $64.6 \pm 4.88$ |
| | EEGNet | $97.86 \pm 0.27$ | $29.14 \pm 1.92$ | $0.38 \pm 0.12$ | $70.36 \pm 1.11$ |
| | CNN | $97.35 \pm 0.48$ | $27.81 \pm 3.39$ | $0.33 \pm 0.03$ | $62.85 \pm 0.79$ |
| | PyramidalCNN | $\mathbf{98.12} \pm 0.18$ | $\mathbf{25.79} \pm 1.25$ | $\mathbf{0.27} \pm 0.08$ | $\mathbf{61.34} \pm 1.28$ |
| | Xception | $97.05 \pm 0.4$ | $28.19 \pm 2.14$ | $0.4 \pm 0.05$ | $63.49 \pm 1.19$ |
| Top23 | InceptionTime | $97.54 \pm 0.91$ | $27.82 \pm 5.48$ | $0.32 \pm 0.03$ | $62.92 \pm 1.66$ |
| | EEGNet | $97.96 \pm 0.24$ | $29.82 \pm 1.31$ | $\mathbf{0.26} \pm 0.01$ | $69.22 \pm 1.21$ |
| | CNN | $98.16 \pm 0.31$ | $25.97 \pm 3.94$ | $0.3 \pm 0.06$ | $62.88 \pm 1.35$ |
| | PyramidalCNN | $\mathbf{98.38} \pm 0.2$ | $\mathbf{24.08} \pm 1.08$ | $0.32 \pm 0.18$ | $\mathbf{61.32} \pm 0.67$ |
| | Xception | $97.96 \pm 0.63$ | $28.35 \pm 2.08$ | $0.37 \pm 0.17$ | $63.2 \pm 1.51$ |
| Top8 | InceptionTime | $97.92 \pm 0.68$ | $26.08 \pm 2.34$ | $0.35 \pm 0.12$ | $65.57 \pm 0.84$ |
| | EEGNet | $98.1 \pm 0.13$ | $29.62 \pm 1.61$ | $0.27 \pm 0.02$ | $71.76 \pm 1.22$ |
| | CNN | $97.92 \pm 0.47$ | $27.87 \pm 3.17$ | $0.32 \pm 0.03$ | $65.37 \pm 1.29$ |
| | PyramidalCNN | $98.12 \pm 0.32$ | $26.21 \pm 1.75$ | $\mathbf{0.26} \pm 0.03$ | $\mathbf{65.33} \pm 1.76$ |
| | Xception | $97.98 \pm 0.24$ | $\mathbf{25.75} \pm 1.03$ | $0.34 \pm 0.07$ | $66.2 \pm 1.66$ |
| Top3 | InceptionTime | $98.0 \pm 0.35$ | $31.94 \pm 3.27$ | $0.36 \pm 0.08$ | $72.28 \pm 1.87$ |
| | EEGNet | $\mathbf{98.46} \pm 0.15$ | $33.64 \pm 1.68$ | $0.38 \pm 0.1$ | $76.12 \pm 0.92$ |
| | CNN | $98.34 \pm 0.24$ | $\mathbf{28.12} \pm 3.92$ | $0.32 \pm 0.03$ | $71.77 \pm 0.75$ |
| | PyramidalCNN | $98.1 \pm 0.67$ | $29.11 \pm 1.98$ | $\mathbf{0.27} \pm 0.04$ | $\mathbf{70.98} \pm 0.59$ |
| | Xception | $97.72 \pm 0.3$ | $30.46 \pm 3.15$ | $0.35 \pm 0.07$ | $74.46 \pm 1.71$ |
| Top2 | InceptionTime | $98.18 \pm 0.49$ | $\mathbf{32.15} \pm 3.31$ | $0.36 \pm 0.05$ | $77.07 \pm 3.68$ |
| | EEGNet | $\mathbf{98.57} \pm 0.11$ | $38.23 \pm 1.55$ | $0.34 \pm 0.01$ | $77.55 \pm 0.89$ |
| | CNN | $97.39 \pm 1.88$ | $33.08 \pm 4.28$ | $0.34 \pm 0.05$ | $75.29 \pm 2.52$ |
| | PyramidalCNN | $98.48 \pm 0.21$ | $33.16 \pm 2.01$ | $\mathbf{0.28} \pm 0.03$ | $\mathbf{75.47} \pm 0.74$ |
| | Xception | $97.84 \pm 0.44$ | $34.94 \pm 2.43$ | $0.47 \pm 0.24$ | $79.21 \pm 1.61$ |

Table 2: **Benchmark.** The performance of all models in EEGEyeNet benchmark for each task and each chosen cluster. The error of left-right task (LR) is measured in accuracy, amplitude and position task is measured in pixels, and angle in radians.

We evaluate all proposed electrode configurations on all deep learning models. The benchmark is run for all the tasks separately on NVIDIA GeForce GTX 1080 Ti. We see that on average equal or better results (in comparison with networks that have access to the full information) can be achieved by just using a fraction of electrodes.

We can observe that in Table 2, running the benchmark with only 23 electrodes the models perform equally and sometimes even better than training the models with all 128 electrodes. This can be since many of the other electrodes do not add any useful information but just noise. This can be seen for example in the Left-Right task where PyramidalCNN achieves a score of 98.46 which is better than any model trained with 128 electrodes. Similar behavior can be seen also in the amplitude task. We can also observe that decreasing the number of electrodes to 8, decreases the performance of the models in the position task, however, the performance of all the other tasks with 8 electrodes still remains equally good compared to the performance with 128 electrodes. Decreasing the number of electrodes to 3 and 2, leads to a decrease in performance on all the tasks except the left-right task which can be decoded with high accuracy with only 2 electrodes. Nonetheless, with only 2-3 electrodes the performance on the amplitude task and the position task decreases significantly.

## 5. Electrode Clustering on Maximally Preprocessed Data

In the previous section, we saw that the most important electrodes for decoding eye movement are the frontal electrodes. More specifically, if we choose 23 frontal electrodes then we achieve the same performance as with the full cap which consists of 128 electrodes.

In this section, we investigate how state-of-the-art preprocessing methods used for neuroscientific applications (maximally preprocessing) affect these results. Maximally preprocessing steps exclude ocular artifacts and include ideally only neurophysiological information. This makes the estimation of gaze position harder, however, it also reveals other insights how brain activity is related to eye movement.

Interestingly, the analysis of the maximmally preprocessed data shows that the electrodes in the occipital part of the brain are also important for inferring gaze direction if one considers only the neurophysiological information. However, even after maximally preprocessing the data, the frontal electrodes can still be used for inferring gaze direction, indicating that the preprocessing might be suboptimal in removing ocular artifacts from the recorded EEG signal. Alternatively one could also speculate that the actual neuronal activity in the frontal electrodes (e.g. frontal eye fields) actually entails information about the eye movement.

### 5.1. Finding Important Electrodes

We use again gradient-based methods to find important electrodes.

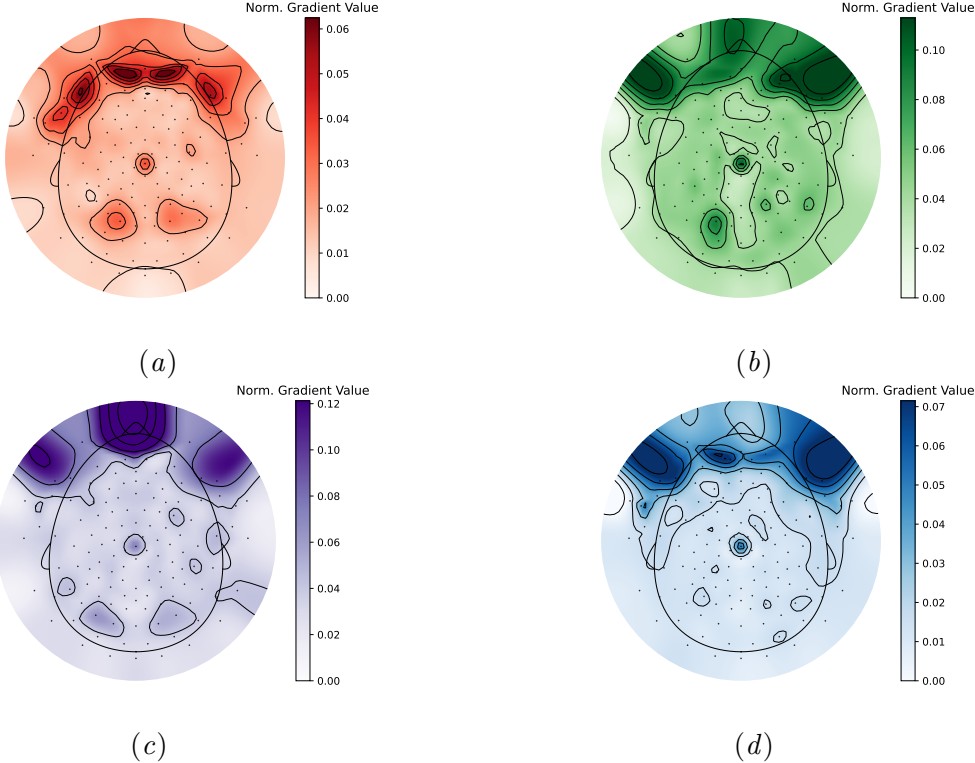

Figure 4: Topoplots for (a) LR Task, (b) Position Task, (c) Amplitude Task and (d) Angular Task. Colors of higher intensity represent a higher importance. The colors are shown on a logarithmic scale. The score is averaged over all five deep learning architectures presented in (Kastrati et al. (2021)).

We observe in Figure 4 again symmetry but, compared to minimally preprocessed data, we can see more sparsity in the distribution of the important brain regions. In particular, in contrast to minimally preprocessed data, we can identify in left-right and direction task an additional important region. Most of the importance is still located in the frontal electrodes, but now the second region of interest around the occipital region can also be identified.

## 5.2. Choice of Electrodes

Due to the symmetry in the topoplots we use the same technique as in the minimally preprocessed data. That is, we rank the electrodes according to the gradient-based analysis (see Figure 5 for the importance of each electrode). With maximally preprocessed data we identified 40 most important electrodes, which achieve almost the same performance compared to the dense 128-electrode cap.

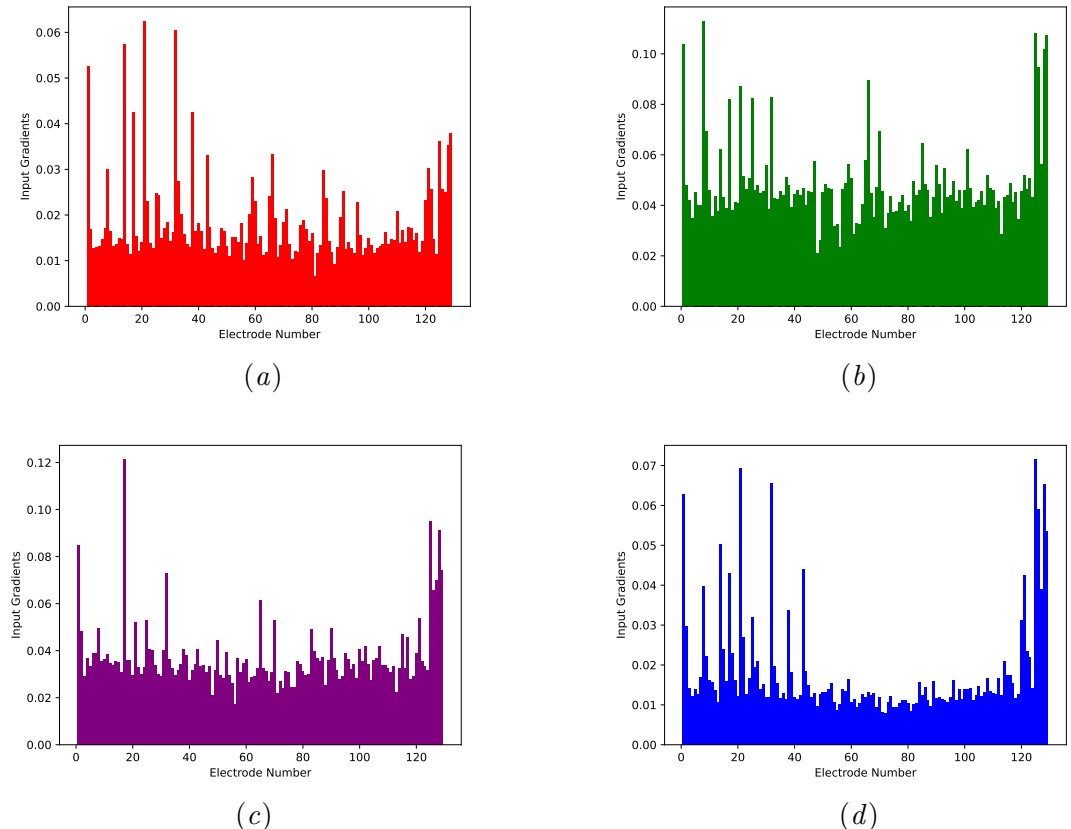

Figure 5: Gradient Based Feature Importance for (a) LR Task, (b) Position Task, (c) Amplitude Task and (d) Angular Task. The normalized gradient results for each electrode and each task from the experiments on the maximally preprocessed data.

## 5.3. Choice of clusters

If we decrease the number of electrodes below 40, then the performance starts decreasing. In this section, we investigate how the performance decreases for several different smaller clusters. Motivated by the fact that there are two different regions in maximally preprocessed data, we distinguish between the following clusters: front, back, front extended, and back extended. The extended version includes more electrodes in each region which results in more stable training. In Figure 6, we show all the main clusters for the best 40 electrodes. In Figure 6 the most important front electrodes are marked with pink color. The front extended cluster is marked with a purple cluster, composed of less important front electrodes. The back cluster is marked in yellow, which is extended with the green cluster (back extended cluster).

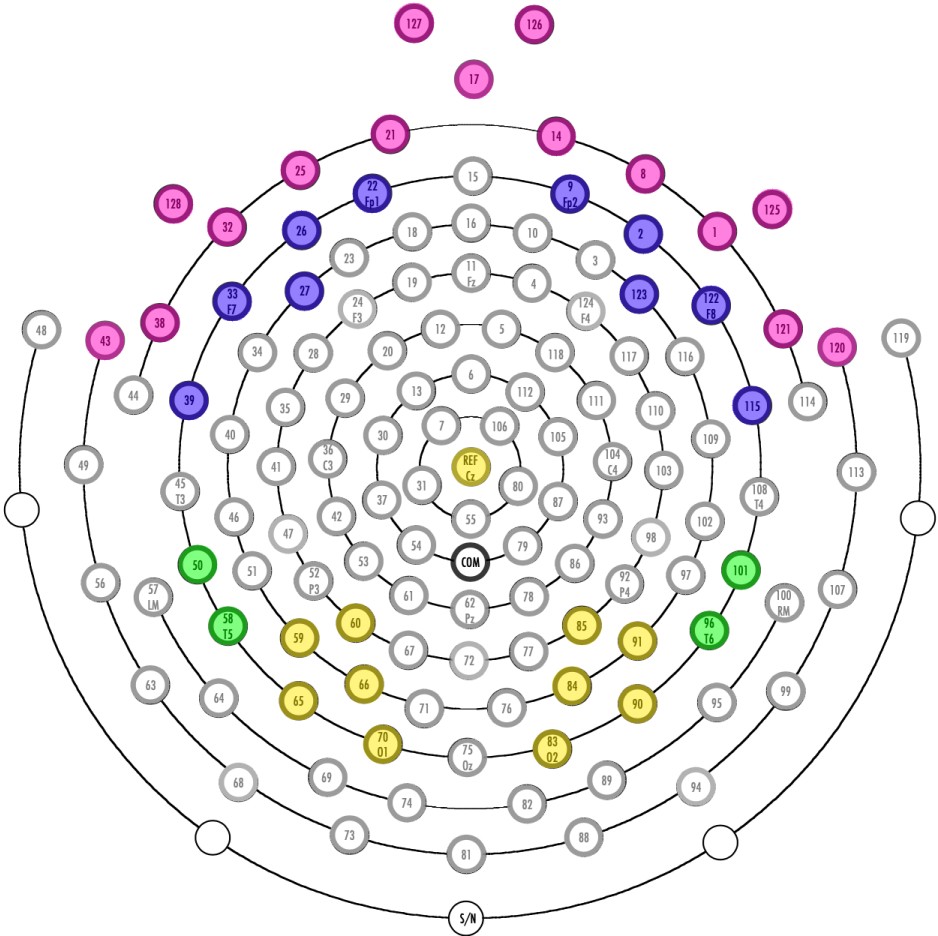

Figure 6: **Electrode Clustering Visualisation.** The best 40 electrodes according to our gradient feature importance method are marked. The pink-colored nodes correspond to the front part. For the extended version, the purple electrodes are added. The same is done for back electrodes. Its base consists of the yellow nodes, the extended version consists of the green ones.

## 5.4. Evaluation

We evaluate all proposed electrode configurations on all deep learning models for the maximally preprocessed data as well. In Table 3, we can see that with only 40 electrodes the models perform equally well and except for LR task, the models perform even better. For example, in the angle task, PyramidalCNN achieves a score of 0.68 if trained with 40 electrodes compared to the score of 0.76 radians if trained with 128 electrodes. Similar behavior can be seen also for the amplitude task. We can also observe that the models achieve competitive performance compared to the trained models with the full cap, also if they are trained with only the front (extended) cluster. Here we can also see for the angle task,

PyramidalCNN trained only on the front extended task achieving a score of 0.67 radian, which is better than all models trained on the full 128-electrode cap.

| Cluster | Model | LR. (%) ↑ | Amp. (mm) ↓ | Ang. (rad) ↓ | Pos. (mm) ↓ |
|---|---|---|---|---|---|
| All (128) | InceptionTime | **93.61** ± 0.67 | 61.42 ± 3.55 | 0.78 ± 0.08 | 125.07 ± 3.43 |
| | EEGNet | 86.31 ± 1.47 | 60.42 ± 0.7 | 1.18 ± 0.05 | **114.06** ± 0.38 |
| | CNN | 90.48 ± 1.87 | 64.31 ± 2.77 | 0.89 ± 0.14 | 122.59 ± 3.81 |
| | PyramidalCNN | 93.08 ± 0.49 | **57.73** ± 1.44 | **0.76** ± 0.09 | 133.59 ± 1.62 |
| | Xception | 89.36 ± 3.01 | 64.2 ± 1.41 | 0.9 ± 0.19 | 125.34 ± 2.79 |
| Top40 | InceptionTime | **93.59** ± 1.05 | 60.42 ± 2.31 | 0.75 ± 0.06 | 122.48 ± 1.98 |
| | EEGNet | 87.95 ± 0.09 | 59.66 ± 0.58 | 1.09 ± 0.03 | **112.9** ± 0.19 |
| | CNN | 90.64 ± 1.09 | 62.38 ± 1.4 | 0.83 ± 0.09 | 119.28 ± 0.62 |
| | PyramidalCNN | 92.01 ± 1.66 | 57.04 ± 0.82 | **0.68** ± 0.02 | 134.6 ± 2.45 |
| | Xception | 92.12 ± 0.8 | **63.0** ± 1.5 | 0.87 ± 0.13 | 125.71 ± 1.96 |
| Front & Back | InceptionTime | **92.6** ± 0.17 | 59.34 ± 3.23 | 0.86 ± 0.19 | 121.48 ± 2.17 |
| | EEGNet | 87.26 ± 0.55 | 60.52 ± 0.68 | 1.12 ± 0.02 | **112.91** ± 0.38 |
| | CNN | 90.99 ± 0.8 | 61.86 ± 2.52 | 0.82 ± 0.03 | 118.94 ± 1.69 |
| | PyramidalCNN | 91.36 ± 1.0 | **58.63** ± 0.47 | **0.75** ± 0.03 | 134.0 ± 2.99 |
| | Xception | 90.07 ± 0.15 | 63.15 ± 3.3 | 0.96 ± 0.11 | 125.17 ± 1.59 |
| Front Ext. | InceptionTime | **92.47** ± 0.81 | **60.72** ± 1.78 | 0.8 ± 0.15 | 124.68 ± 2.12 |
| | EEGNet | 83.94 ± 1.41 | 63.35 ± 0.92 | 1.08 ± 0.01 | **113.83** ± 0.36 |
| | CNN | 91.42 ± 0.48 | 69.73 ± 5.59 | 0.8 ± 0.03 | 119.23 ± 0.92 |
| | PyramidalCNN | 91.28 ± 1.69 | 60.94 ± 1.98 | **0.67** ± 0.03 | 135.0 ± 2.27 |
| | Xception | 91.76 ± 0.25 | 66.74 ± 3.45 | 0.81 ± 0.09 | 124.06 ± 1.94 |
| Front | InceptionTime | **91.68** ± 0.19 | **61.31** ± 2.5 | 0.91 ± 0.15 | 123.05 ± 0.9 |
| | EEGNet | 83.33 ± 0.63 | 63.25 ± 0.27 | 1.12 ± 0.02 | **114.19** ± 0.39 |
| | CNN | 90.57 ± 0.82 | 61.84 ± 2.28 | 0.85 ± 0.04 | 120.67 ± 3.23 |
| | PyramidalCNN | 89.54 ± 1.8 | 61.36 ± 1.2 | **0.78** ± 0.02 | 135.15 ± 3.52 |
| | Xceptio | 91.04 ± 0.69 | 62.6 ± 2.98 | 0.85 ± 0.07 | 125.44 ± 2.74 |
| Back Ext. | InceptionTime | 71.42 ± 2.48 | 78.46 ± 3.53 | **1.52** ± 0.08 | 130.57 ± 1.05 |
| | EEGNet | **74.32** ± 0.43 | **67.93** ± 0.19 | 1.81 ± 0.01 | **119.88** ± 0.1 |
| | CNN | 71.39 ± 1.02 | 75.04 ± 1.81 | 1.54 ± 0.07 | 127.8 ± 1.55 |
| | PyramidalCNN | 70.91 ± 1.32 | 70.85 ± 0.58 | 1.56 ± 0.03 | 145.75 ± 2.04 |
| | Xception | 70.67 ± 1.29 | 81.64 ± 1.15 | 1.66 ± 0.06 | 132.16 ± 0.66 |
| Back | InceptionTime | 69.21 ± 1.85 | 75.45 ± 2.48 | 1.67 ± 0.08 | 127.42 ± 1.63 |
| | EEGNet | **72.98** ± 0.49 | **68.72** ± 0.37 | 1.78 ± 0.07 | **119.87** ± 0.18 |
| | CNN | 70.02 ± 0.63 | 79.54 ± 8.81 | **1.54** ± 0.04 | 126.45 ± 1.93 |
| | PyramidalCNN | 67.88 ± 3.94 | 73.03 ± 2.25 | 1.61 ± 0.02 | 145.6 ± 1.69 |
| | Xception | 68.19 ± 2.03 | 81.93 ± 2.57 | 1.7 ± 0.05 | 132.81 ± 2.23 |

Table 3: **Benchmark.** The performance of all models in EEGEyeNet benchmark for each task and each chosen cluster for maximally preprocessed data. The error of left-right task (LR) is measured in accuracy, amplitude and position task is measured in pixels, and angle is measured in radians.

The position task for maximally preprocessed data is difficult for all the models even with the full cap, and as stated in (Kastrati et al. (2021)) it is not clear how good this task can be solved with only neurophysiological data. Finally, compared to the minimally preprocessed data, we can also see eye movement information can be decoded with only the electrodes on the occipital regions of the scalp, however, the performance decreases significantly. For instance, the best model trained with frontal electrodes achieves an accuracy of 92.58%, whereas the best performing model trained with the electrodes on the occipital part achieves an accuracy of 74.53%. If only the occipital electrodes are used the performance on the other tasks decreases significantly. This can be seen for the angle task where the performance of the models changes from 0.67 to 1.55 radians. This error is close to the naive baselines reported in Kastrati et al. (2021), showing the difficulty in learning more complex eye movements other than left-right classification solely from occipital electrodes.

## 6. Bandpassing

The final analysis of the impact of preprocessing consists of bandpassing our data before training.

### 6.1. Choice of frequency bands

We decided to focus our attention on a limited number of intervals, roughly based on historically pre-defined frequency bands (Newson and Thiagarajan (2019)). We define four frequency intervals: Delta : 1-4Hz, Theta : 4-8Hz, Alpha : 8-13Hz and Beta : 13-32Hz. The raw EEG measurements are bandpass for each subject before preparation of the input data, to allow precise frequency selection before reducing the signals to 1-second intervals to feed our model. This bandpassing is performed on both the maximally and minimally preprocessed data, before training respectively on each band.

### 6.2. Results

We present the results below, including two frequency intervals obtained by merging frequency intervals. We report the average results of the EEGEyeNet benchmark for the left-right task. In Figure 7, we observe an important drop in accuracy for each of our bandpass datasets, with expectedly higher results when combined.

Interestingly, the maximally preprocessed dataset seems to contain most of its helpful information in the higher frequencies, namely the 13-32Hz dataset, with an accuracy above the combination of the three other intervals. It seems to be the contrary for our minimally preprocessed dataset, with a high accuracy in 4-8Hz frequency band and 1-13Hz frequency band prediction. We see no significant improvement from the inclusion of the 13-32Hz interval. We hypothesize that the significant front electrodes' importance observed during clustering on the minimally preprocessed data contains mainly sub-13Hz signals used for prediction. A potential reason for this is that saccades happen only every 200-300ms. Additionally, the fact that 13-32Hz frequency band is important for maximally preprocessed data indicates that actual neuronal activity is used for decoding.

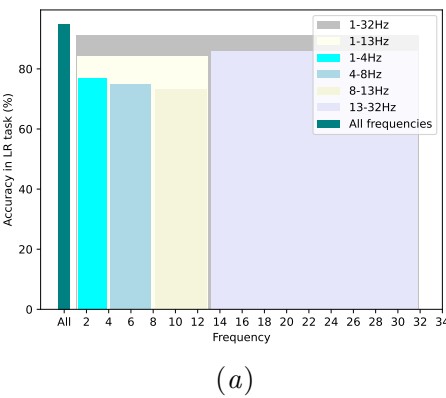
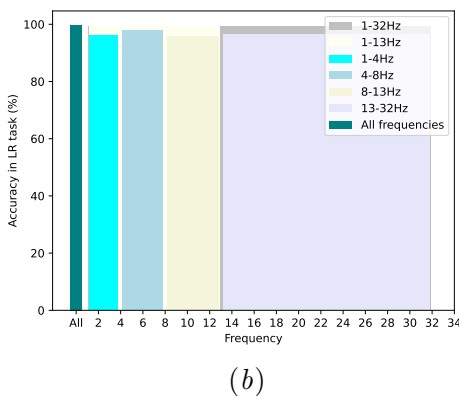

$$(a) \qquad\qquad\qquad (b)$$

Figure 7: **Bandpass analysis.** The relation between the frequency intervals and the performance of the models in the LR task for the (a) maximally preprocessed data and (b) minimally preprocessed data.

## 7. Conclusion

The study's major finding shows that minimally preprocessed data used for prediction contains most of its information for the task in a limited number of electrodes in the frontal part. Therefore, by reducing their number to one-quarter of its original number, we decreased the input data size significantly and stabilized the prediction results with special cases where the performance even increases. Moreover, the short fit of our data and the high stability of the results might suggest the possibility of reducing the model's complexity without accuracy loss.

It is interesting to note that also, for the maximally preprocessed dataset, the frontal electrodes are most important for gaze prediction. Since this dataset was treated with a non-brain artifactual source components removal based on the automatic classification result as provided by Independent Component Analysis, we can speculate the signal stems from the frontal eye field area, which plays an essential role in controlling visual attention and eye movements (Armstrong et al. (2009)). The oculomotor artefact removal in the maximally preprocessed dataset makes the gaze estimation position task more challenging. Nevertheless, this dataset contains mostly neurophysiological information and reveals other insights into how brain activity relates to eye movement. In particular, it is interesting to note how the second region of interest, located at the occipital part of the head, was important for inferring gaze direction. Since, most likely, the electrodes located in this part of the head are not influenced by any residual oculomotor noise, we can conclude that this signal contains information measured in the region of the visual cortex, revealing how neurophysiological brain activity is related to eye movement.

Finally, we analyzed the impact of bandpassing our data before training. As expected, the low frequencies were most significant for the minimally preprocessed dataset, as they are related to ocular artefacts. However, the maximally preprocessed dataset revealed a different pattern, showing that the 13-32Hz frequencies contained the most meaningful information

for our tasks. A current limitation in this work is that the bandpass analysis is performed only on the left-right task and for limited windows of frequency bands. Extensions of this work call for a fine-grained spectral analysis and how the performance changes across different eye movement patterns (other than left-right).

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
