# OpenReview forum: "Electrode Clustering and Bandpass Analysis of EEG Data for Gaze Estimation"
_NeurIPS.cc/2022/Workshop/GMML — Gaze Meets ML 2022 Oral_

### Official Review · Reviewer_nhaM · 2022-10-15
**EEG lead placement analysis for gaze tracking**

**Rating:** 6
**Confidence:** 3

**Review:**

This work is primarily in the area of EEG analysis, with contributions in determining the optimal number and location of electrodes, for the purpose of eye gaze. The eye gaze tracking data is used for training.

Eye tracking using EEG has been long studied by HCI researchers.

The methodology is sound. Authors have limited computational contribution, mostly repeating previous work. The new additions are in EEG analysis and electrode placement.

---

### Official Review · Reviewer_dm5p · 2022-10-17
**Gaze decoding with EEG**

**Rating:** 7
**Confidence:** 5

**Review:**

This paper examines gaze signals within EEG. This work is a cool blend of state-of-the-art computer science deep learning techniques with neuroscience, and EEG-based eye-tracking, and these results could be a powerful tool for computer science and neuroscience. The authors show that eye movements can be reconstituted with a small number of electrodes and specific frequency bands with a high degree of accuracy, even after preprocessing.

Comments:
* One methodological consideration: Given the relative importance of the reference electrode (vs. nearby electrodes), the interpolation after referencing might be problematic. My suggestion would be to mask out the reference electrode and not use it in the models.
* The authors interpret evidence that 13-32 hz frequency band as evidence that actual neural activity is used for decoding. However, I do not necessarily think that 13-32 hz is interpreted as neural activity, and an alternative conclusion could be that the preprocessing is most adept at removing ocular artifacts in lower frequencies. My suggestion would be to remove this interpretation of 13-32 hz activity, but I do think it is important that preprocessing changes which frequencies might support classification.
* In general, the authors found reasonably consistent results with different architectures, and did not examine the differences between architectures that would give rise to different results. They could consider combining the results across all architectures, or focusing on PyramidalCNN in the main text
* I am not convinced from these results that the frontal decoding can be attributed, even speculatively, to FEF, especially given the electrode positions combined with the notorious challenges of the inverse problem . The authors could revise that statement to refer generally to eye gaze information within the frontal regions of the brain.
* An overarching conceptual challenge to this paper is whether they are training on ocular or neural activity from EEG. The authors make inroads on this problem, but ultimately the dataset, though expansive, is limited by the lack of EOG activity. Comparing neural vs. eog activity is outsideof the scope of this project, and the authors do not need to address it in this work. However, the authors could suggest future directions for this work, including whether datasets that include EOG data (in addition to EEG and eye gaze) could be important for understanding the nature of EEG based gaze decoding.

Minor:
* The rationale for which results are bolded in the tables should be added to the captions

---

### Official Review · Reviewer_CjB2 · 2022-10-19
**Solid workshop paper**

**Rating:** 9
**Confidence:** 3

**Review:**


The paper presents a study on Electroencephalography (EEG) based gaze estimation.
The authors investigates how to reduce the cost of EEG caps.
Specifically they expriment reducing the number of electrodesin the EEG cap and demonstrates that for the purposes of EEG-based eye tracking it only leads to a small, affordable drop in performance.
Good workshop paper.

---

### Meta-Review · Area_Chair_8gyg · 2022-10-20

**Recommendation:** Accept (Oral)
**Confidence:** 4

**Metareview:**

This work studies the use of EEG-based gaze estimation. Reviewers indicated the promising experimental findings of being able to capture eye movements with a smaller number of electrodes in the EEG cap, and have also provided additional feedback for improvements (see Reviewer dm5p list of comments). An immediate suggestion for improvement would be to incorporate related work from the HCI literature, as pointed out by Reviewer nhaM. Overall, the paper has received positive feedback that suggests acceptance.

---

### Decision · Program_Chairs · 2022-10-20

Accept (Oral)